# Human-Centered Design of a Collaborative Robotic System for the Shoe-Polishing Process

Giorgia Chiriatti, Marianna Ciccarelli [ID], Matteo Forlini [ID], Melania Franchini, Giacomo Palmieri * [ID], Alessandra Papetti [ID] and Michele Germani

Department of Industrial Engineering and Mathematical Sciences, Università Politecnica delle Marche, 60121 Ancona, Italy
* Correspondence: g.palmieri@univpm.it

**Abstract:** Demand for automated processes in the manufacturing industry is now shifting toward flexible, human-centered systems that combine productivity and high product quality, thus combining the advantages of automated and robotic systems with the high-value-added skills of operators and craftsmen. This trend is even more crucial for small and medium-sized enterprises operating in the "Made in Italy" fashion industry. The paper presents the study, simulation, and preliminary testing of a collaborative robotic system for shoe polishing that can reduce manual labor by limiting it to the finishing stage of the process, where the aesthetic result is fully achieved, with a benefit also in terms of ergonomics for the operator. The influence of process parameters and design solutions are discussed by presenting preliminary test results and providing hints for future developments.

**Keywords:** human–robot collaboration; fashion industry; leather shoe polishing; Industry 4.0

## 1. Introduction

The demand for automated processes in the manufacturing industry is nowadays inevitable for enterprises that want to be competitive in the market. The high-precision automation technology improves production efficiency and reduces manual tasks with a high probability of work errors [1]. In recent years, collaborative robots (cobots) are deeply changing the manufacturing and production processes to guarantee flexibility in production and high levels of safety [2]. The industrial manipulators are going to be replaced with cobots to accomplish complex collaborative tasks sharing the workspace with the operator [3]. Among the various objectives, the human–robot collaboration aims to improve the psychophysical well-being of the operator and reduce his cognitive overload, guaranteeing safety requirements [4]. This aspect is even more important in sectors where the manual component plays a fundamental role such as handicrafts. For example, in the footwear industry, most of the manufacturing processes are handmade and only some processes are dedicated to specialized machinery such as cementing and cutting operations [5]; while the use of robots is generally dedicated to ordinary repetitive tasks, such as pick and place [6], depositing glue on the sole [7], and sole robotic grasping during assembling operations [8]. The use of robots is limited for the high variability that characterizes the shoe manufacturing process (from the choice of the kind of shoe to the materials that composes it). To cover the large variety of shoe types and styles, the production has to be flexible and adapted quickly to any modifications. Therefore, the machines must be easily reconfigurable for the achievement of functional and fashion goals of the industry. The main distinctive functions of a shoe are comfort, protection, and style which bring the shoe industry to reach both functional and fashion goals.

Among the most critical aspects of the leather shoe supply chain, especially if aimed at the luxury market, there are:

- Low productivity, which is an intrinsic characteristic of craft operations, but partly due to the limited ability of companies to combine traditions and technological innovations. Generally, this sector uses static and mature technologies and automation remains limited [9];
- A lack of flexibility that collides with the need for reduced time-to-market, high levels of customization, and micro-lots (or single pairs) management [10];
- Medium/high ergonomic risks with a higher probability of occurrence of musculoskeletal disorders for operators [11];
- Difficulty in managing and passing on the knowledge of experienced craftsmen, which results in training courses that are often too long and inadequate [12].

An essential process for the production of luxury shoes is the polishing operation, which is mostly handmade. The result of good leather polishing is determined by the operator's skills, where the force applied on the surface is experienced-based determined. The operator applies the polish several times by performing a series of circular movements with variable frequency and pressure according to the area of the shoe to be polished. This operation takes about 15–20 min per shoe. It often involves the operator assuming incorrect postures, as well as repetitive movements. These factors make this process not ergonomic.

To obtain accurate shoe polishing and overcome the limits of traditional operation, an advanced robotic solution to automate the process is proposed. The major goal of polishing automation is to improve time efficiency together with surface quality. During the operation, a proper contact force between the robot and the shoe surface must be maintained and robots are required to have a high degree of compliance to control the final output. In the past, several studies have been oriented toward the design of robots that can work in contact with the workpiece with force interaction, such as low-mobility parallel robots [13,14] that can offer high precision and stiffness; however, the complexity of the kinematics and small workspace have limited their use in very specific applications. Advances in sensors and control have led to new types of robots, such as collaborative robots (cobots), in which force sensing and advanced software and control tools can be leveraged to achieve precise manufacturing operations. In the present application, the cobot is exploited to lead several tools for polishing different types of workpieces with several controllable parameters, such as the contact force, the rotational speed of the tool, and the feed rate [15]. Force control can be implemented by either passive compliance control or active force control [16,17]. In the passive compliance control, the polishing tool is deformed based on the external force through means of some compliant mechanical components that mainly absorb energy when contact occurs with the surfaces. Active control, instead, is an adopted control strategy that actively controls the polishing force on the surface according to the feedback information on sensors [18]. The real-time normal force between the polishing tool and surface must remain constant; meanwhile, the system controls the position and posture of the polishing tool. Robotic polishing systems are used widely in many manufacturing companies including the automobile, aircraft, iron, and steel industries where abrasive brushes are applied to smooth or do the final touches of the working surface without affecting its geometrical dimensions. According to the literature, most of the studies related to automatic polishing using collaborative robotics are conducted on complex surfaces for finishing operations [19–21], while the literature on their specific use for the shoe-polishing process is lacking. The paper suggests the innovative study of shoe polishing using a collaborative robot in the footwear industry, by addressing the following main challenges:

- To ensure the precision required for handling and processing a "delicate" and deformable object, such as leather footwear, to achieve high-quality polished leather. Force control must be very accurate so that the level of surface finish is comparable to that of hand-worked leather, avoiding even minor defects. Consequently, standard surface-finishing techniques and tools will have to be optimized and adapted to the specific case.

- To formalize the artisans' knowledge in polishing recipes, which vary according to models, materials, and colors.
- To define the modalities of the human–robot collaboration so that the cell respects the safety requirements, and the process exploits synergistically the automation advantages and the added value given by the human operator. The aim is to facilitate the operator's work, reduce his workload, and preserve his health.

The rest of the paper is organized as follows. Section 2 describes the polishing process of leather shoes and its formalization to then be automated. Section 3 illustrates the methods for realizing the collaborative work cell, its elements, its layout, and the tool path simulation. Section 4 presents the process simulation of human–robot interaction and discusses the preliminary results of laboratory tests. Finally, concluding remarks and future works are summarized.

## 2. The Polishing Process of Leather Shoes for the Luxury Market

To formalize the knowledge in terms of polishing rules based on shoe characteristics, the process was analyzed in detail by observing the operators themselves and collecting their considerations on the key aspects of the process. This study led to the identification of eleven main tasks, which are shown in Figure 1.

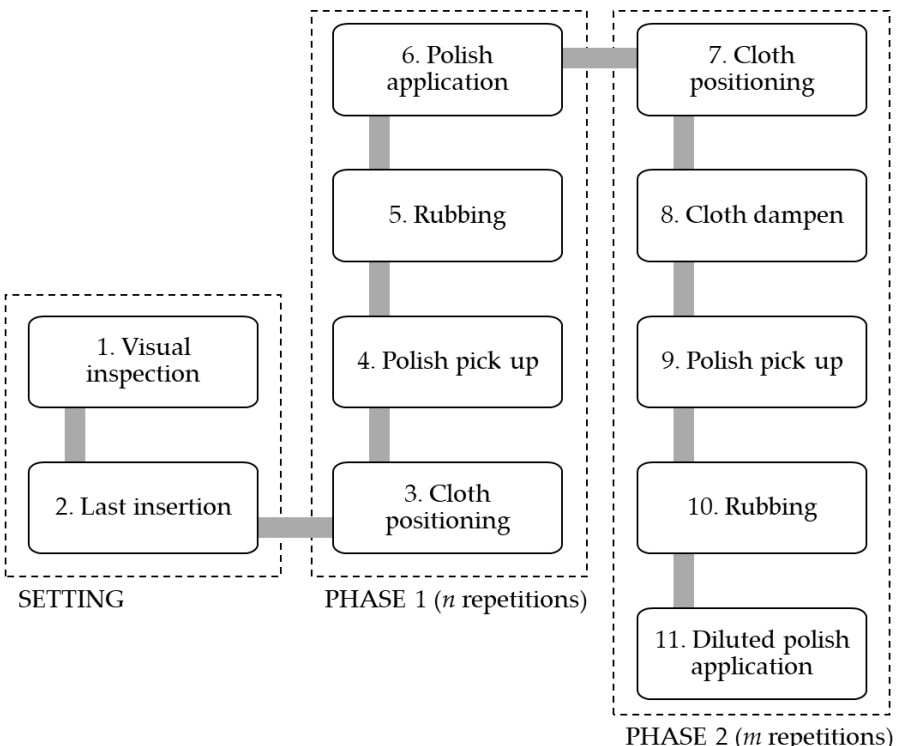

**Figure 1.** Polishing process of leather shoes.

The visual inspection of the pair of shoes allows the identification of the peculiarities to be managed, for example, the base color that will determine which and how many polishes must be used, any differences between the two shoes that involve a different execution of the process on the same to uniform the final appearance, etc. The insertion of the last inside the shoe allows for the obtaining of the right resistance to the pressure exerted by the operator.

The first phase consists of four main tasks. The operator places the cloth on the fingers to have the right sensitivity when picking up and applying the polish. The pick-up of the polish takes place through the cloth and exerting pressure with the fingers inside the container. One or more polishes can be used depending on the color of the shoe. The rubbing on white tape allows for the avoiding of overdosing and to distribute the polish

evenly on the fingers. This step is extremely important to avoid staining the shoe on first contact, which could affect the success of the entire process. The application of the polish on the shoe always takes place starting from the toe, the counter, or the lower edge since these are the points where the nuance will be more accentuated. The other areas will be gradually reached to achieve the desired shade.

In the second phase, the same tasks are performed, but the polish is diluted (task 8).

Phase 1 and Phase 2 are repeated n and m times, respectively, where n and m are defined from time to time based on the considerations that emerged from task 1 and the goodness of the polishing process.

The shoe was divided into different areas based on (i) the movements made by the operator (e.g., linear, circular) and (ii) the geometries of tools needed to better reach all parts of the shoe model.

Subsequently, a further subdivision of the shoe into areas was carried out in order to define the polishing rules. This subdivision arises from the observation of different polishing processes performed by the operators and considering the number of passes carried out on the various areas and the pressure exerted. As already mentioned, the darkest areas will be the toe, the counter, and the bottom edge. Each polish cycle starts from these areas and then gradually proceeds towards the center of the shoe.

In Figure 2, an exemplifying subdivision is shown due to confidentiality reasons.

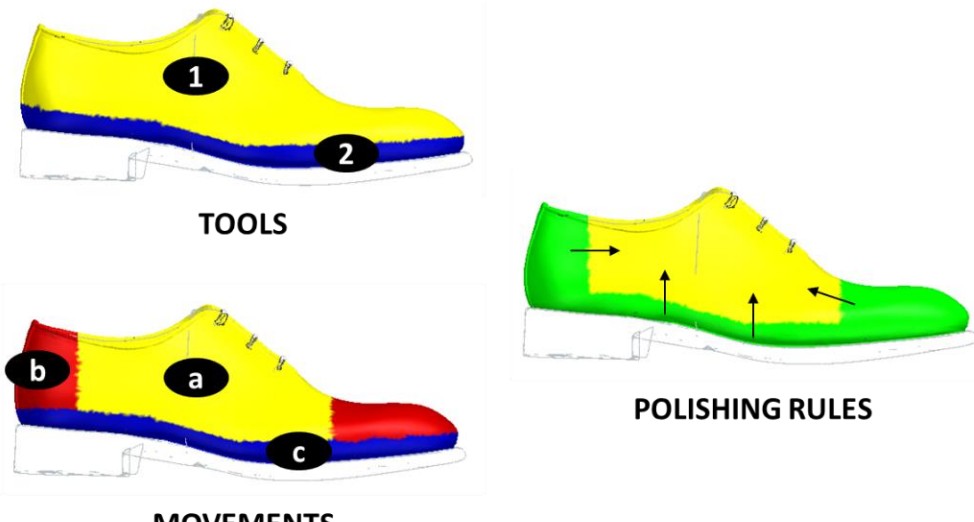

**Figure 2.** Subdivision of the shoe into areas based on tools, movements, and polishing rules.

Starting from these considerations, standard polishing cycles can be defined as a sequence of contiguous areas. These cycles are then customized by users based on the characteristics of the shoe. Specific parameters (repetitions, pressure, speed, etc.) can be set on the dedicated user interface.

The operator's exposure to ergonomic risks has been objectively assessed by using the XSens® MVN inertial motion capture system. The operator was equipped with 18 Xsens MTw (Wireless Motion Tracker, Xsens, Enschede, The Netherlands) while performing the polishing activities. Figure 3 shows the resulting risk score according to the RULA (Rapid Upper Limb Assessment) method. The main risk factors are related to the wrist and neck position. The repetitiveness of the movements also plays a key role, and its reduction would determine significant benefits: very high risk can be reduced from 8% to 2% and medium risk from 58% to 11%. The HRC (Human–Robot Collaboration) would enable this opportunity by automating the first phase of the process. The ergonomic design of the workstation could instead improve the postures assumed by the operator.

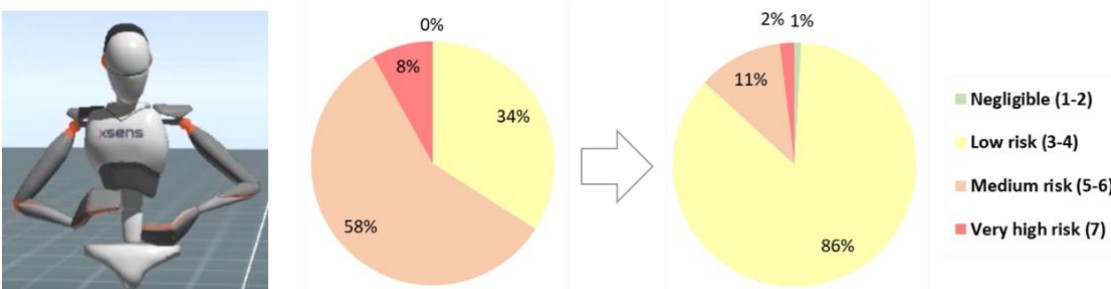

**Figure 3.** Ergonomic evaluation of as – is process and potential benefits obtainable with HRC.

## 3. Functional Description of the Robotic Cell

### 3.1. Robotic Work Cell

The entire shoe polishing process is designed as a sequence of operations performed by the robot and the operator in close cooperation. The primary polishing operation is performed by the robot under the supervision of the operator who, based on his experience, decides when the automatic process should stop or possibly requires the robot to retrace only a section of the path. After the preliminary robotic polishing, the operator performs light manual finishing. While the operator performs the finishing operation, the robot is asked to start a new polish on a different shoe.

The proposed robotic cell is composed by:

- a 6-DOF collaborative manipulator (UR5e) by Universal Robots;
- an aluminum chassis;
- a worktop;
- a shoe-locking system;
- a polish dispenser;
- an internal supporting system of the shoe.

The cobot selected for the application is suitable for medium-capacity operations, with 5 kg of payload and a workspace of 850 mm radius. In order to reach all the points of the shoe, the manipulator is mounted inverted on the aluminum structure to execute the polishing from above (Figure 4). The area around the shoe is free and the operator can work without space limitations caused by the robot.

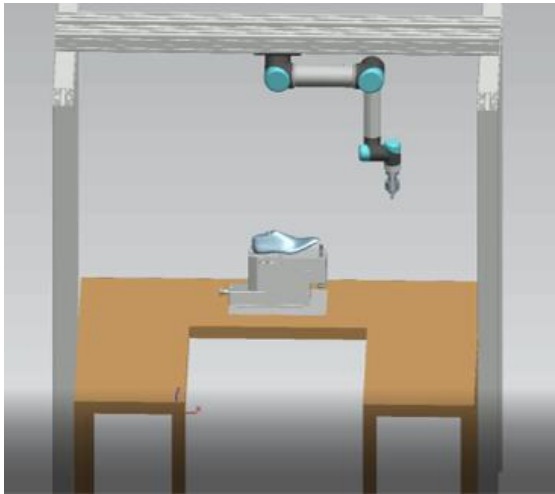

**Figure 4.** UR5e mounted on the profile structure.

The workstation is composed of five zones, all easily reachable by the operator (Figure 5):

- Zone 1 is the storage area for shoes to be polished
- Zone 2 is the storage area for end products

- Zone 3 is the robot workspace with the dispenser and the shoe-locking system
- Zone 4 is for manual processing by the operator
- Zone 5 is the area scanned by a laser sensor for speed and separation monitoring.

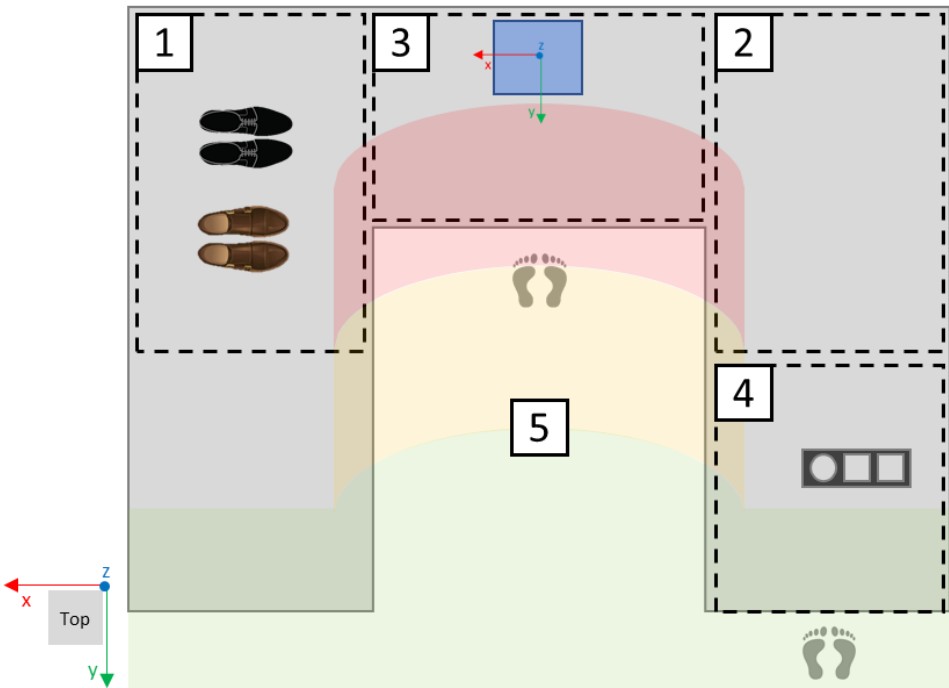

**Figure 5.** Workstation zones.

The layout of zones 1, 2, and 3 is designed following the principles of the golden zone (The golden zone in ergonomics refers to the area nearest to the core of your body between your shoulders and knees. Even better, the Occupational Safety and Health Administration (OSHA) recommends sticking to an area between the mid-thigh and mid-chest to allow employees to lift with even more ease.) and strike zone (In a person, the strike zone is between the knuckle and shoulder height. This area has been identified by ergonomists as the optimal zone for lifting, handling and carrying material because workers can move their hands freely without too much reach or bend.); these areas must be privileged, in terms of ergonomics, for the use of materials and tools to carry out activities. The table in zone 4 is adjustable in height to adapt to the different anthropometric characteristics of the operators and reduce postural risks.

The introduction of a laser scanner sensor located under the central part of the robot's work area allows it to detect the presence and distance of the operator and instantly order the robot to slow down (orange area) or stop (red area). The manipulator will resume the process with normal speed only when the operator is outside the orange area again. It is not excluded that a full collaboration between the robot and the operator can be reached in the speeds of the robot are reduced, making possible a manual intervention during the automatic work done by the robot without any protective stop.

A dedicated shoe-locking system is designed in order to guarantee repeatability in the polishing process performed by the robot or the operator. In this way, the reference frame can be precisely located at the shoe toe. The system is easily adaptable to different sizes of shoes thanks to an adjustable posterior component that, according to the shoe size, can fix the heel with a self-adapting mechanism without causing any damage to the footwear. Technical details for the shoe-locking system are not reported due to a confidentiality agreement.

The dispenser for the polish is a fixed system that lets the tool of the robot autonomously operate for the entire process. The system allows a rapid change of color and the homogenization of the polish. It is composed of three essential components, respec-

tively, used for holding the polish can through a magnet (1), for applying water through an absorbent felt (2), and for homogenizing the polish through a steel plan (3) shown in Figure 6, in which details are not included due to confidentiality reasons.

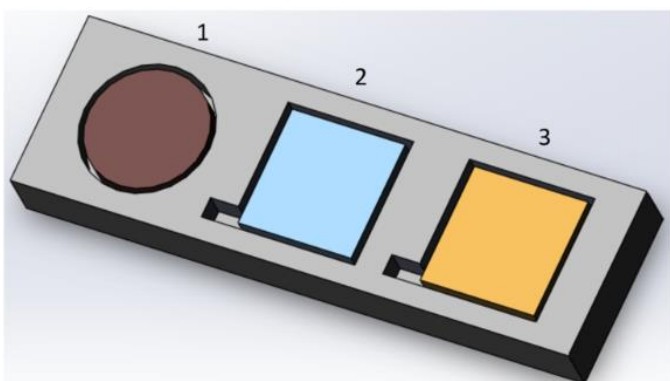

**Figure 6.** Polish dispenser.

The robotic cell is also equipped with a system used to support the inner part of the shoe, consisting of a cotton-covered polyurethane sock and actuators that allow the expansion of the sock by compressed air at 5 bar. In this way, the shoe is able to bear the forces applied by the robot with limited local deformations.

### *3.2. Tool Path*

Defining the procedural path is a crucial aspect that will determine the precision and accuracy of the process. It was decided to implement it with software that allows the automation of the manual process by simulating and verifying machine operations. CAM NX software offers solutions for machine tool programming, post-processing, and production simulation, so it can be chosen as a development platform for the process design under study.

Among the different operations given by NX Manufacturing the most similar to the polishing process is the "Multi-axis-deposition", which allows the creation and simulation of the deposit of a polish layer on the surface. During the operation it is possible to change different parameters, to adapt the tool to the shoe geometry and reach every surface, opting for a zig-zag path combined with the tool movement.

In the fashion industry, the traditional methods of product design often entail the realization of a 2D paper model of the shoe upper without the support of CAD or similar technologies. For this aim, the reverse engineering approach was adopted to regain design data on company products. Two alternative strategies have been investigated, that imply the 3D scan of the shoe and the 3D scan of the last, respectively. The former entailed very complex and time-consuming post-processing; therefore, the second approach was followed. The latter entails the application of a shim to simulate the shoe upper and to carry out the correction in the areas in which the geometry is significantly different. According to the results obtained from the analysis of the shoe areas, described in the past chapter, to boost the process accuracy, the CAD model has been divided into sections, and then it polishing cycles are defined.

The general rule followed, for every polishing cycle, is the begging of the polishing from the blue area, defining the distance of 5 mm between the lines at the start of the process to gradually increase until 10 mm in the upper areas. The variation of the distance between the lines, which will be defined through experimental tests, gives the possibility to obtain the shading effect (Figure 7).

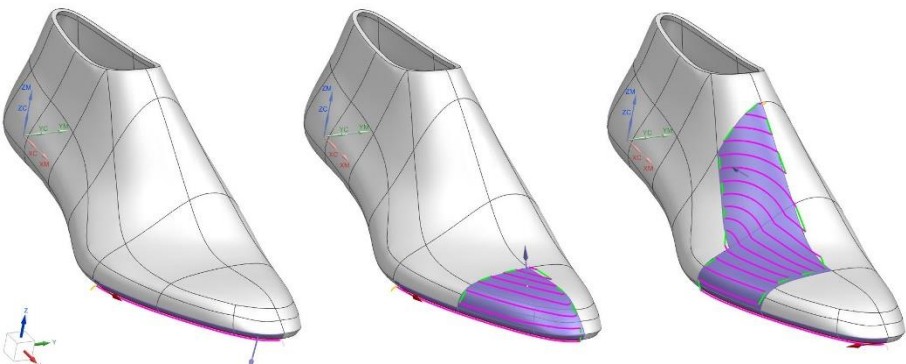

**Figure 7.** Tool path simulation on shoe last.

### 3.3. Polishing Tool

The polishing tool is designed to simulate the movement of the operator's hand. In the manual operation, the operator places a cloth on the fingers and collects and applies the polish on the shoe (Figure 1, PHASE 1). To replicate the movement of human fingers, two possible kinematic solutions are proposed. In the first solution, the tool rotates about its fixed axis. The second solution is based on a planetary motion in which the tool rotates about an axis which moves on a circular trajectory. Referring to Figure 8, the ratio between the angular velocities of the planet gear ($\omega_p$) and carrier ($\omega_c$) can be determined as:

$$\frac{\omega_p}{\omega_c} = 1 - \frac{z_r}{z_p} \tag{1}$$

where $z_r$ and $z_p$ are, respectively, the number of teeth the ring and planet gears. The tool head is in this case fixed to a planet gear, whereas the motor is coupled to the carrier.

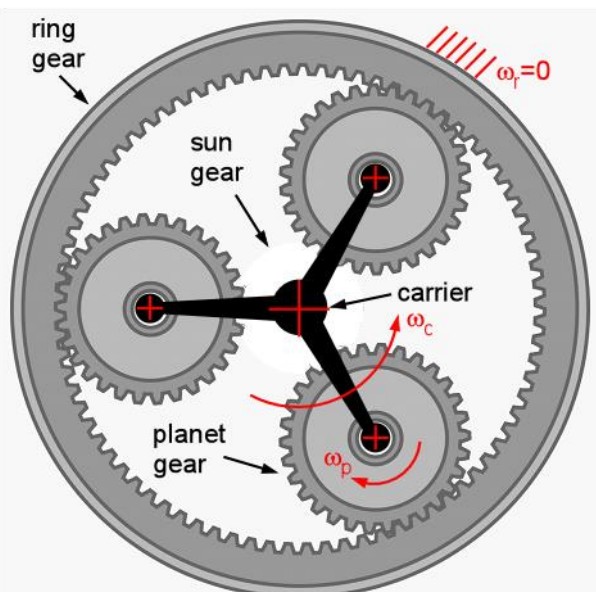

**Figure 8.** Planetary mechanism.

The mechanical design of the polishing tool, shown in Figure 9, consists of a flange and a tube-shaped outer support that holds the inner elements, shown in Figure 10 for the case of fixed-axis rotation and Figure 11 for the case of planetary motion. The mechanical design was conceived to easily switch from one solution to the other, thus changing the type of tool movement: by removing a single cover of the outer support, it is possible to act on the coupling (c) shown in Figure 10, replacing the tool head in a very short time.

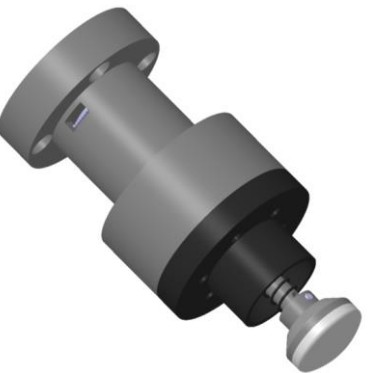

**Figure 9.** Exterior view of the polishing tool with 30 mm diameter backing pad.

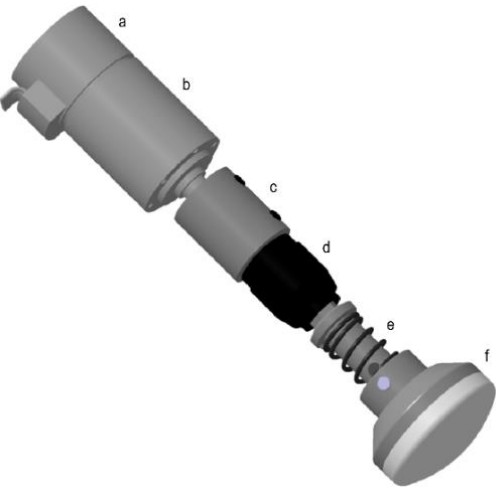

**Figure 10.** Internal components for the fixed-axis rotation design: (a) motor, (b) gear, (c) coupling joint, (d) quick release coupling, (e) spring, (f) 30 mm diameter backing pad.

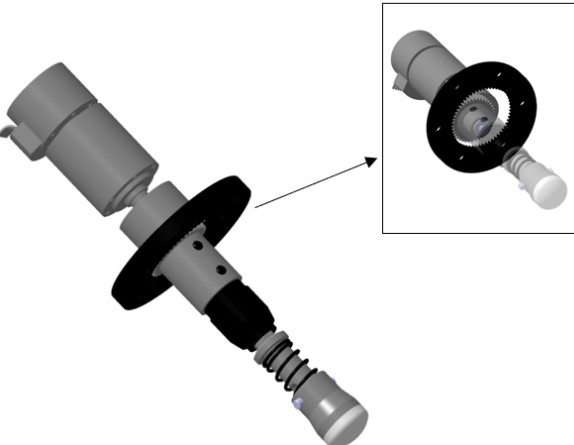

**Figure 11.** Internal components for the fixed-axis rotation design with a 15 mm diameter backing pad.

The tool head consists of a spring that mimics the compliance of human fingers and a backing pad that can have different sizes, such as 15 mm and 30 mm in diameter. This allows for easy adaptation to different parts of the shoe: the tool with the smaller diameter is suitable for polishing narrow surfaces, such as the area 2d shown in Figure 2, while the larger backing pad is more suitable for larger surfaces, such as the area 1c shown in Figure 2.

The actuation system consists of a DC motor with a nominal voltage of 24 V, whose size in terms of power and torque was determined based on estimated process parameters, such as polishing speed, coefficient of friction, and the estimated operator's normal force applied to the leather. A view of the assembled prototype of the polishing tool is shown in Figure 12.

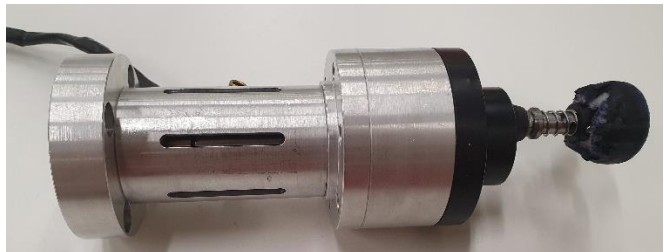

**Figure 12.** Prototype of the polishing tool.

## 4. Simulation and Preliminary Tests

In this section, the simulations and the preliminary tests to evaluate the feasibility and the effectiveness of the robotic polishing system are described.

### 4.1. Simulation

The human–robot collaboration is simulated on Tecnomatix Process Simulation software to provide the feasibility of the process. The simulation involves operations appropriately sequenced by the robot and by the operator described in the following steps (Figure 13):

1.  The operator stands in the green area of zone 5 (a).
2.  The operator enters zone 5 and instantly the robot slows down (orange area in Figure 6). The operator enters the red area to take one shoe from zone 1 and places it on the locking system in zone 3 (b, c). The robot automatically stops.
3.  The operator leaves the red/orange area of zone 5 and waits for the end of the robotic polishing (d, e). The robot operates with its normal execution velocity.
4.  After the end of the robotic polishing, the operator enters the red area of zone 5, takes the shoe from the lock system, and replaces it with a new one to be polished (f, g).
5.  The operator places the shoe polished by the robot in zone 4 to carry out manual refining work (h).
6.  The operator places the finished product in zone 2 and waits for the end of robotic polishing always in the green area of zone 5 (i).

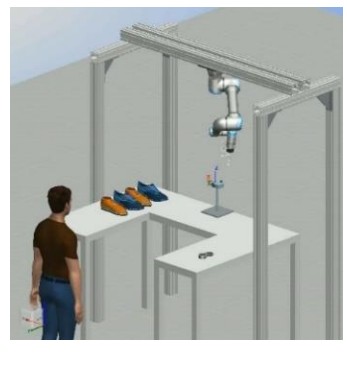 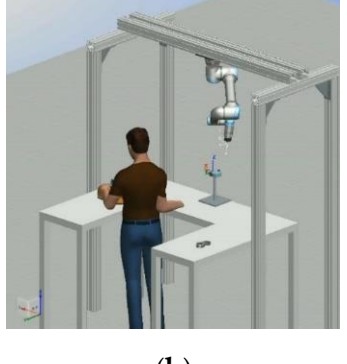 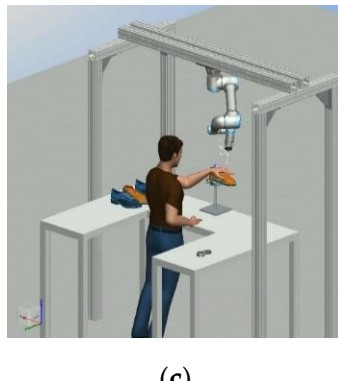

(**a**)                        (**b**)                        (**c**)

**Figure 13.** *Cont.*

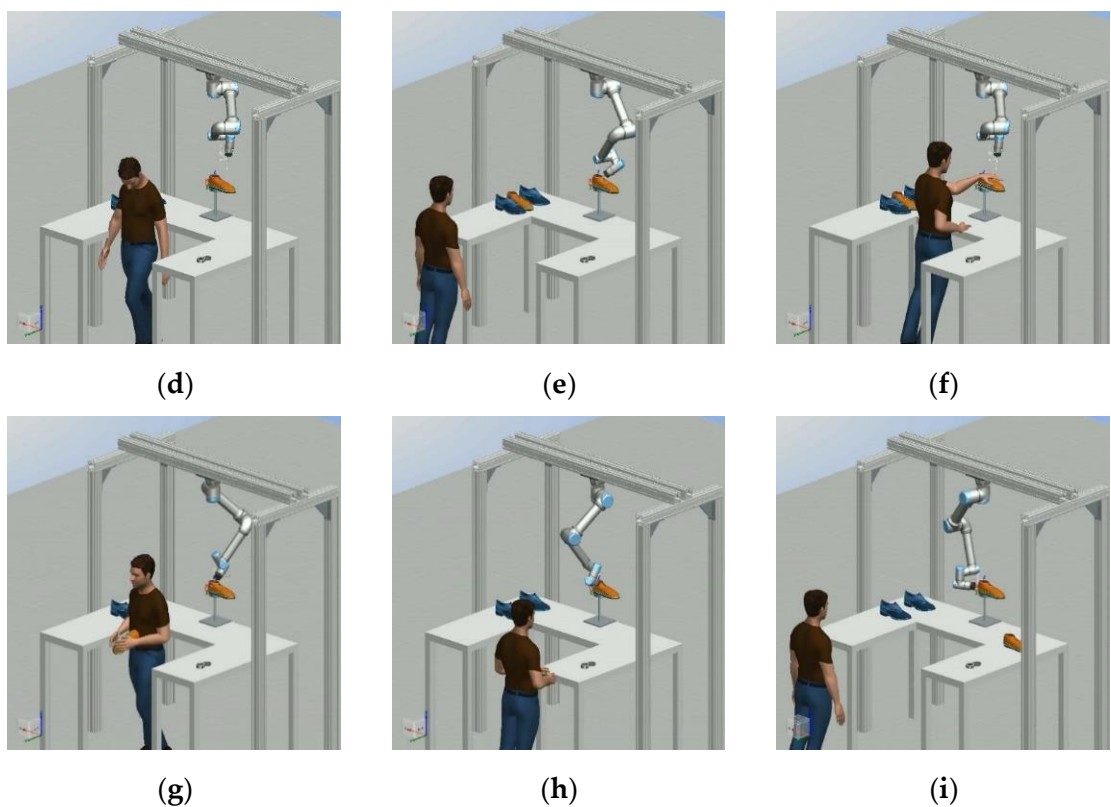

(**d**)         (**e**)         (**f**)

(**g**)         (**h**)         (**i**)

**Figure 13.** Simulation of the polishing process.

These operations are repeated until the shoes are ended. Therefore, the operator can still interrupt the robotic polishing process at any time and, in case of necessity, act manually on the shoe.

To assess the operator's safety during the robotic process and to avoid the fact that the elbow of the robot works near the operator's eyes, the height of the aluminum-profile structure is set at 2040 mm and the height of the table at 1050 mm. In this condition, the robot works away from singularities and the subjects, simulated with 50th and 95th percentile of the body measures, can easily reach the footwear.

The robot executes the polishing process by following the path described in Section 3.2. The path is post-processed in NX through the cls file in which all the points available for the trajectory are listed. These points are imported in the Tecnomatix environment with the z-axis perpendicular to the surface of the shoe (Figure 14a). The Tool Center Point (TCP) of the robot must be in continuous contact with the surface of the shoe (Figure 14b) with the z-axis always perpendicular to the surface.

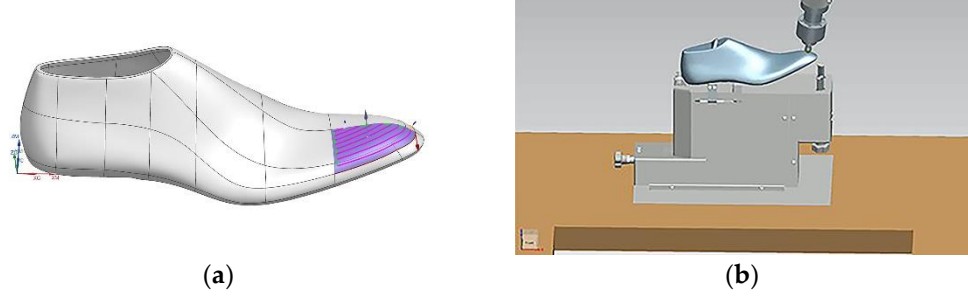

(**a**)         (**b**)

**Figure 14.** Simulation of the path: (**a**) Trajectory on the shoe surface simulated in NX; (**b**) Simulation of the robot TCP in contact to the shoe surface in Tecnomatix environment.

### 4.2. Preliminary Tests

The first experimental tests are performed on flat leather to evaluate the input required for robotic processing to obtain the first polished leather. Only after identifying the required inputs, the process will be performed on the footwear, as described in the paper. These tests involve the polishing of flat leathers with the UR5e manipulator. The leathers used are divided into two different areas (brown and blue) as shown in Figure 15, while the tool is the same described in Section 3.3. In this experimental section, the two kinematic solutions for the movimentation of the tool are tested to analyze and compare the result of the robotic polishing. Since the leather is plane and fixed on the table, the robot is mounted directly on the workbench, and the dispenser for picking up and homogenizing the polish is located on the right side of the robot. The set-up is shown in Figure 16 and the robotic operations (picking-up, homogenization, and polishing) are reported in Figure 17.

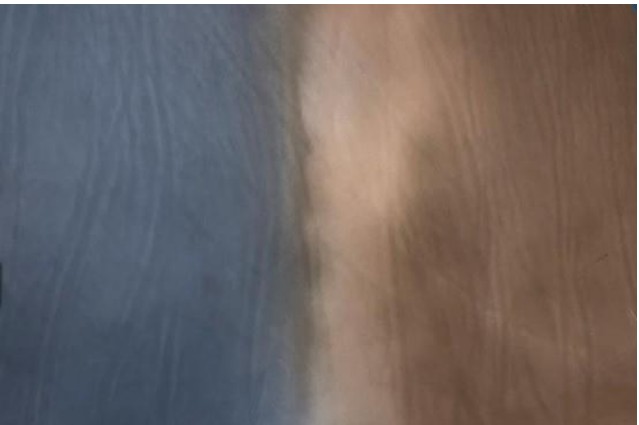

**Figure 15.** Flat leather before polishing.

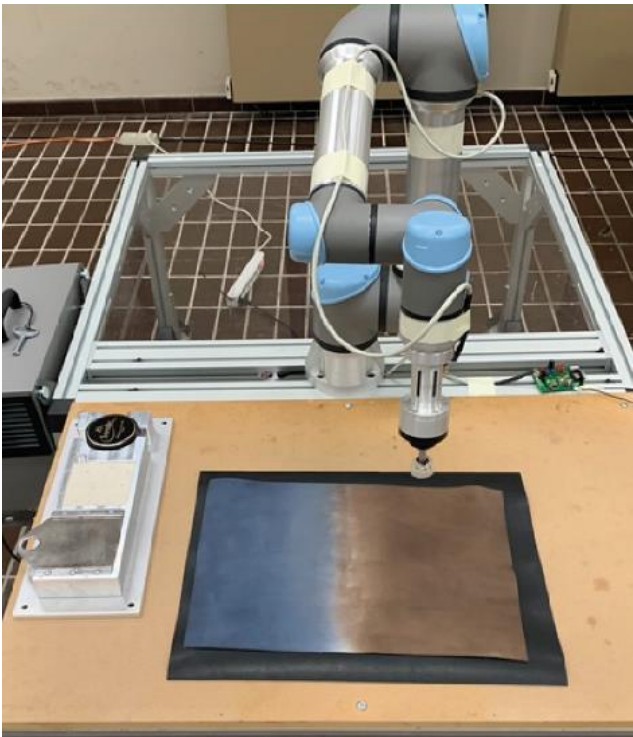

**Figure 16.** Robotic polishing station.

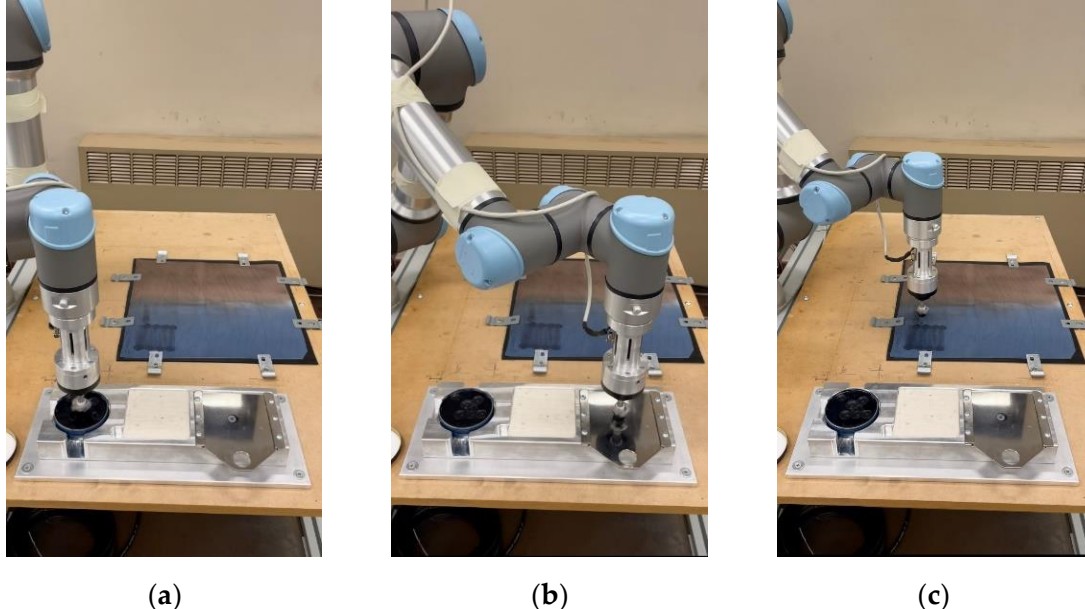

**Figure 17.** Three phases of the process: (**a**) picking up of the polish; (**b**) homogenization of the polish; (**c**) polishing.

The operation of polishing involves the definition of the robotic path and the force control applied on the surface points. The robotic path, illustrated in Figure 18, is initially composed of four lines equally spaced by step p (a) in which each line is performed by the robot from left to right and vice versa. Then, the robot follows a continuous path from left to right, moving with an offset (ds) from the first path line (b). In the end, the robot executes the same continuous path starting from the endpoint in the opposite direction (c).

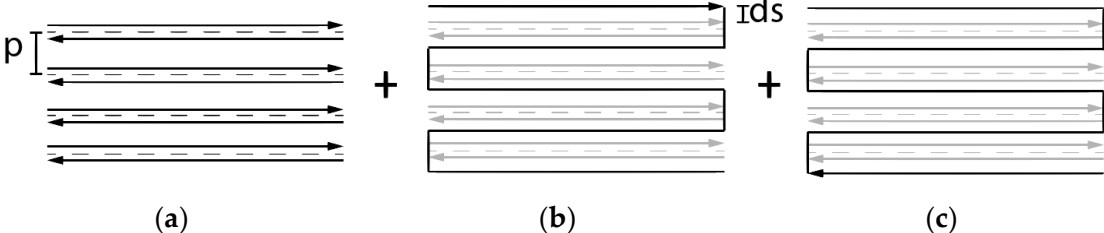

**Figure 18.** Polishing path: (**a**) Four lines separated by the step (p); (**b**) First continuous path with offset ds; (**c**) Second continuous path with offset ds.

The robot applies a normal force along the entire area to be polished. The force is measured by the integrated load cell on the end flange of the manipulator and controlled by standard software tools (UR PolyScope software platform). Based on the different stages of the path, the maximum value of the force changes. In particular, the force applied on the single lines (a) is greater than those on the continuous path (b) and (c) because during the first path the robot must be able to leave on the leather a large amount of polish. Through the continuous path, instead, the application of the normal force is lighter to better distribute the polish on the surface and make it more uniform without leaving unpolished points. The force applied on the single lines ranges from a minimum of 2 N to a maximum of 12 N, while on the continuous path the maximum normal force is set at 8 N. Figures 19 and 20 show the leather before (a) and after (b) the robotic polishing for both blue and brown types with the planetary motion and the fixed-axis motion, respectively. After the first complete phase of the polishing, the leather gets the initial shine and color necessary for the subsequent treatments (Figure 1, PHASE 2). In particular, in the planetary motion, the robotic procedure that generates the best result involves the picking-up and

the homogenizing phases before polishing each line of the path, while the polishing of the continuous path is executed without having picked up the fresh polish to better expand the polish accumulated on the leather in the previous step. A different approach is followed in polishing with the fixed-axis tool motion: the picking-up and the homogenizing phases are always executed before the polishing phase and the whole procedure is repeated two times.

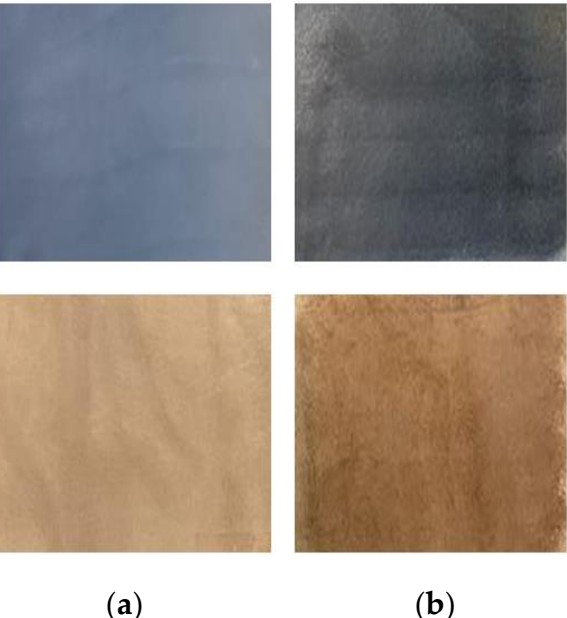

**Figure 19.** Leather before and after polishing obtained with the planetary motion and a backing pad of 30 mm diameter: (**a**) Leather before polishing; (**b**) 60 mm × 60 mm area of polished leather.

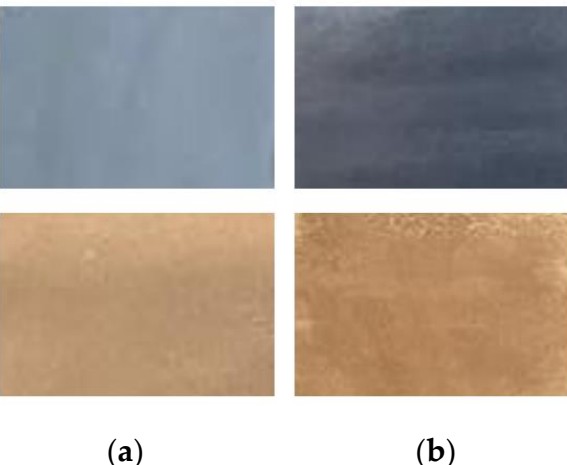

**Figure 20.** Leather before and after polishing obtained with the fixed-axis tool motion and a backing pad of 15 mm diameter: (**a**) Leather before polishing; (**b**) 50 mm × 40 mm area of polished leather.

These first results are obtained by setting the parameters reported in Tables 1 and 2. The parameters linked to the area to be polished are the height (h), width (l), the step (p), and the offset (ds), already illustrated in Figure 18. The kinematic parameters are the velocities, the accelerations of the robot during different stages of the process, and the tool velocity. The robot velocities are different during the process, with ($v_{polish}$) and without ($v_{unpolish}$) the fresh polish on the end-effector, and in the service velocity ($v_{service}$) with which the robot performs the picking-up and the homogenization of the fresh polish. The tool velocity ($v_{tool}$) corresponds to the rotation speed of the tool head. The fast acceleration ($a_{fast}$) is the acceleration of the robot during the service tasks while the slow acceleration

($a_{slow}$) is applied during the polishing phase. The force-related inputs are divided into the force that the robot exerts on the leather during the polishing operation of the first path (Figure 18a) ($F_1$), the force applied during the second and third path (Figure 18b,c) ($F_2$), the force of the robot when picks-up the polish ($F_{pick-up}$) and the force of the robot during the homogenizing phase ($F_h$). Then, three input parameters related to the time the robot takes to collect the polish ($t_{pick-up}$), the homogenization time ($t_h$), and the time it takes to reach the leather before starting the polishing step ($t_{sleep}$) are established.

**Table 1.** Inputs of the process related to the results shown in Figure 19.

| Parameters | Blue Leather | Brown Leather |
|:---:|:---:|:---:|
| h [m] | 0.06 | 0.06 |
| l [m] | 0.06 | 0.06 |
| p [m] | 0.01 | 0.01 |
| ds [m] | 0.008 | 0.008 |
| $v_{polish}$ [m/s] | 0.07 | 0.07 |
| $v_{unpolish}$ [m/s] | 0.5 | 0.5 |
| $v_{service}$ [m/s] | 1 | 1 |
| $v_{tool}$ [rpm] | 5000 | 5000 |
| $a_{slow}$ [m/s$^2$] | 0.4 | 0.4 |
| $a_{fast}$ [m/s$^2$] | 1.4 | 1.4 |
| $F_1$ [N] | 12 | 3 |
| $F_2$ [N] | 10 | 2 |
| $F_{pick-up}$ [N] | 2 | 2 |
| $F_h$ [N] | 2 | 1 |
| $t_{pick-up}$ [s] | 5 | 5 |
| $t_h$ [s] | 4 | 4 |
| $t_{sleep}$ [s] | 0.09 | 0.08 |

**Table 2.** Inputs of the process related to the results shown in Figure 20.

| Parameters | Blue Leather | Brown Leather |
|:---:|:---:|:---:|
| h [m] | 0.04 | 0.04 |
| l [m] | 0.05 | 0.05 |
| p [m] | 0.008 | 0.008 |
| ds [m] | 0.005 | 0.005 |
| $v_{polish}$ [m/s] | 0.1 | 0.1 |
| $v_{unpolish}$ [m/s] | 0.5 | 0.5 |
| $v_{service}$ [m/s] | 1 | 1 |
| $v_{tool}$ [rpm] | 5000 | 5000 |
| $a_{slow}$ [m/s$^2$] | 0.1 | 0.1 |
| $a_{fast}$ [m/s$^2$] | 1.4 | 1.4 |
| $F_1$ [N] | 3 | 4 |
| $F_2$ [N] | 2.5 | 2.5 |
| $F_{pick-up}$ [N] | 3 | 3 |
| $F_h$ [N] | 3 | 2.5 |
| $t_{pick-up}$ [s] | 5 | 5 |
| $t_h$ [s] | 4 | 4 |
| $t_{sleep}$ [s] | 0.05 | 0.05 |

## 5. Conclusions

The polishing operation for the production of luxury shoes is an essential aspect of the shoe manufacturing process. This operation is mostly handmade and takes about 15–20 min per shoe where the result of good leather polishing is determined only by the operator's skills. The aim of this study is to propose an advanced robotic solution through collaborative robots to automate the process and overcome the limits of traditional operation, including ergonomic risk reduction. The first feasibility study shows that the robot, properly mounted on the aluminum-profile structure, perfectly reaches the whole

shoe surface and works in total safety for the operator. Moreover, the addition of a laser scanner sensor adds more safety for the operator and a reduced working modality for the robot.

The feasibility of the study is verified by the simulations. The post-processing of the shoe through the CAM software allowed the simulation of a polish deposition path on the shoe surface that is suitable for each shape. After performing the entire path of the shoe, the simulation is inserted in the Tecnomatix Process Simulation to have a complete simulation of the activities of the operator and the robot. This allows for the checking of the reliability of the entire process and the reachability of all the points of the shoe for polishing. The purpose of this study is also to verify the output of robotic polishing with the application of the designed tools. Tests are done on several flat pieces of leather (always blue and brown types) aimed to find the input parameters needed for good results. In the test done with the planetary motion with a backing pad of 30 mm diameter (Figure 19), the leather appears well-polished and the polish is uniformly distributed on the surface. The quantity of polish on the leather is suitable for the first polishing phase. On the other hand, in the test done with the fixed-axis motion with a backing pad of 15 mm diameter (Figure 20), the result is less evident due to the low presence of polish on the leather. However, the leather is polished with less polish on the surface and the motion around the vertical axis does not allow a uniform distribution of the polish. To achieve an acceptable result with this type of tool, the complete robotic procedure must be repeated twice. Moreover, a substantial difference between the two motions of the tool is the force applied by the robot to polish the blue leather ($F_1$) which in the planetary motion is four times higher than the other motion.

In analyzing the phases of the robotic procedure, some problems emerged starting from the consistency of the polish in the can. The first layer of the polish is stiffer than the ones underneath and the comparison of the force of picking-up ($F_{pick-up}$) among tests is impeded. A solution is to remove the first layer of polish before performing the tests and have the polish always with the same consistency. In addition, the surface used to homogenize the polish on the tool presents some problems as the polish is well distributed on the center of the tool head, but it is accumulated at its edge. A future solution will be to use a convex contact surface to better distribute the polish on the tool. Further studies will apply these strategies directly to the shoe surface.

**Author Contributions:** Conceptualization, G.P. and A.P.; investigation, M.C., M.F. (Matteo Forlini), M.F. (Melania Franchini) and G.C.; supervision, M.G. All authors have read and agreed to the published version of the manuscript.

**Funding:** This research received no external funding.

**Data Availability Statement:** Not applicable.

**Conflicts of Interest:** The authors declare no conflict of interest.

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
