# Peer review of "Human-Centered Design of a Collaborative Robotic System for the Shoe-Polishing Process"

_machines, doi:10.3390/machines10111082_

Round 1
Reviewer 1 Report (New Reviewer)
Dear authors,
My suggestions for your article are presented below.
1. The explanation of abbreviations such as SME should be given in the first place.
2. It would be good to emphasize the contribution to the literature and the importance of your work.
3. The apostrophe sign of the "Figure" word on line 187 should be fixed.
4. How the force is measured?
5. On line 198, what does the meaning of the golden zone and strike zone? It is recommended to explain briefly, and be shown in related figure if possible.
6. Equation number 1 must be right-justified.
7. Figure 21 mentioned on lines 415 and 456 does not exist.
8. On line 461, it is stated that the polishing procedure is approximately 15-20 minutes per shoe, does the applied method reduce the production time?
With my best regards
Author Response
Please see the attachment

Reviewer 2 Report (New Reviewer)
1. The mechanism design of the shoe polishing robotic systems is very clear, but the key sensing and control algorithms are not stated, such as the force control law and signal estimation.
2. I wonder if the human has the right to change the polishing missions of the robotic systems in practice.
3. The procedures of the polishing platform design are could be explicit improved currently, since the joint parameters and initial states of the robots are not considered in detail.
Author Response
Please see the attachment

Reviewer 3 Report (New Reviewer)
The paper under review presents the simulation and preliminary experimentation of a robotic collaborative system for shoe polishing capable of reducing manual work and also advantages in terms of ergonomics for the operator.
The topic is more current than ever and it is presented in a clear and exhaustive manner in the procedure and in the results obtained.
I recommend a minor revision of the text in English and a little improve the references.
The work is worthy of a prompt publication of the article in the Journal Machines.
Author Response
Please see the attachment

This manuscript is a resubmission of an earlier submission. The following is a list of the peer review reports and author responses from that submission.
Round 1
Reviewer 1 Report
The current manuscript describes the polishing operation of leather for the production of luxury shoes. It describes, with details, the regular process to perform the polishing and the required synergy and cooperation between the user and the machine. The writing is well done and easy to follow. However, it has a more descriptive and informative rather than scientific approach. The results, although still initial as the authors themselves claim, are indeed promising and relevant but the final deliverable lacks of deeper literature review presentation, clarification of the approached methodology and further analysis and discussions of its results.
For these reasons, I believe the submission is not ready for publication in a scientific journal, despite strong potential to be future published with the final results.
Other points to be improved in this work, in my point of view, are to follow:
-
No references for methods and devices are given (lines 144 to 147).
-
Word-breaking problems at the end of the line several times (e.g. pol-ishes). This is probably a latex hyphenation problem.
-
No clarifications about the novelty being brought by the authors regarding technological innovation are provided and no comparison to other methods is presented.
-
No discussions about where actually the human-robot collaboration happens, or investigated methods to do so, are presented.
-
Parts of the “Result” section are still about describing the experimental setup. Maybe opening a section exclusively for it would benefit the manuscript’s structure. Or at least naming it “experiments and results”.
Finally, an interesting and promising improvement to the work would be apply it in a real world and compare the RULA outcomes to check how much the proposal has in fact prevented ergonomic risks. Otherwise, this claim seems unsupported at the end. It would be nice to analyse how much time would be spared of the process by using the authors’ proposition instead of the regular mode without the robot support.